# Convergence of SVGD in KL divergence via Approximate gradient flow

## Abstract

This study investigates the convergence of Stein variational gradient descent (SVGD), which is used to approximate a target distribution based on a gradient flow on the space of probability distributions. The existing studies mainly focus on the convergence in the kernel Stein discrepancy, which doesn't imply weak convergence in many practical settings. To address this issue, we propose to introduce a novel analytical approach called $(\epsilon, \delta)$-*approximate gradient flow*, extending conventional concepts of approximation error for the Wasserstein gradient. With this approach, we show the sub-linear convergence of SVGD in Kullback–Leibler divergence under the discrete-time and infinite particle settings. Finally, we validate our theoretical findings through several numerical experiments.

## 1 Introduction

Sampling from an unnormalized target distribution, such as posterior distribution in Bayesian inference, is a fundamental problem in machine learning. The mainstream approaches for obtaining such samples is using Markov Chain Monte Carlo (MCMC) methods (Hastings, 1970; Welling & Teh, 2011) or approximating the target distribution by variational inference (VI) (Jordan et al., 1999; Blei et al., 2017). While MCMC provides guarantees of producing asymptotically unbiased samples from the target density, it tends to be computationally intensive (Robert & Casella, 2004). On the other hand, VI achieves a computationally efficient approximation of the target distribution through stochastic optimization under a simpler alternative distribution; however, it does not come with a guarantee of obtaining unbiased samples (Blei et al., 2017).

To alleviate such sample bias while maintaining computational efficiency of VI as much as possible, Liu & Wang (2016) introduced *Stein variational gradient descent* (SVGD), which allows the direct approximation of the target distribution without the need for alternative distributions. SVGD iteratively updates correlated samples, referred to as *particles*, by minimizing the Kullback–Leibler (KL) divergence between a distribution of particles and the target distribution through a gradient flow on the space of probability distributions. Since the Wasserstein gradient is intractable in practice, SVGD approximates it through a kernel method.

On the theoretical front, analysis has been actively conducted ever since Liu (2017) elucidated the asymptotic behavior of SVGD from the perspective of gradient flow within the reproducing kernel Hilbert space (RKHS). Korba et al. (2020) showed sub-linear convergence in kernel Stein discrepancy (KSD) under infinite particles assuming that KSD at each step is bounded. Salim et al. (2022) contributed a proof of sub-linear convergence in KSD without the necessity of bounded KSD assuming that the target distribution satisfies $T_1$ inequality (Villani, 2008), and Sun et al. (2023) provided the proofs of this convergence property by relaxing the smoothness assumption of the target distribution. A common thread in these analyses is seeing SVGD's update rule as the approximation of the Wasserstein gradient in the RKHS and showing that the KL divergence to target distribution monotonically decreases like gradient descent. Beyond the infinite particle setting, Shi & Mackey (2023) has recently shown that the SVGD with $n$ finite particles and an appropriate step size converges in KSD at the $\mathcal{O}(1/\sqrt{\log \log n})$ order if the target distribution is sub-Gaussian with a Lipschitz score.

However, the convergence analysis in terms of KSD is insufficient to understand the weak convergence property of SVGD because the convergence in KSD holds under highly restrictive conditions for the kernel and the target distribution under practical settings as shown by Gorham & Mackey (2017). This fact underscores the importance of conducting convergence analysis using criteria

other than KSD to provide more realistic guarantees for the obtained particles. A natural candidate for the criterion is the KL divergence itself, which is the objective function of SVGD. Recently, Liu et al. (2023) showed that SVGD with finite particles achieves linear convergence in KL divergence under a very limited setting where the target distribution is Gaussian. However, the analytical approach presented in previous studies makes it difficult to conduct convergence analysis based on KL divergence in a more global setting. The reason for this lies in the fact that while the logarithmic Sobolev inequality (LSI) (Gross, 1975) is typically employed to show the linear convergence in KL divergence for a gradient flow in the space of probability distributions (Villani, 2008), it becomes apparent that the inequality similar to the LSI (see Eq. (7)) does not hold in practical settings (Duncan et al., 2023) when considering SVGD as a gradient flow in the RKHS.

In this study, we introduce a novel analytical approach that allows us to circumvent the aforementioned issue. A key idea in our analysis is to consider SVGD as an *approximation* of the gradient flow in the space of probability distributions, as opposed to the conventional analytical approach that views SVGD as a gradient flow in the RKHS. To express the degree of this approximation, we introduce a new concept called $(\epsilon, \delta)$-*approximate gradient flow*, which extends the concept of approximation error widely used in the gradient estimation context such as score gradient estimation (Lee et al., 2022; 2023) and particle-based VI (Liu et al., 2019; Dong et al., 2022).

With our concept, we offer new insights into the convergence of SVGD in the settings of discrete-time and an infinite number of particles. We first analyze the degree of the approximation error $\{\epsilon, \delta\}$ between the Wasserstein gradient of the KL divergence and the update rule in SVGD by focusing on spectral decomposition specified via a kernel function. With this approximation error analysis, we show that SVGD exhibits sub-linear convergence in the KL divergence for the first time, to the best of our knowledge. At last, we conduct a numerical study to examine the convergence behavior of SVGD across various metrics and validate the soundness of our theoretical findings.

## 2 PRELIMINARIES

Random variables are denoted by capital letters like $X$, while deterministic values are denoted by lowercase letters like $x$. The Euclidean inner product and distance are expressed as $\langle \cdot, \cdot \rangle$ and $\| \cdot \|$, respectively. Let $\mathcal{X} = \mathbb{R}^d$ and let $C^l(\mathcal{X}, \mathcal{Y})$ be the space of $l$ continuously differentiable functions from $\mathcal{X}$ to a Hilbert space $\mathcal{Y}$. We abbreviate $C^l(\mathcal{X}, \mathbb{R})$ as $C^l(\mathcal{X})$. The set of smooth functions with compact support is expressed as $C_c^\infty(\mathcal{X})$. If $\phi \in C^1(\mathcal{X})$, its gradient is $\nabla \phi$. For $\phi \in C^1(\mathcal{X}, \mathcal{X})$, the Jacobian is represented as $J\phi(x)$, a $d \times d$ matrix at each point $x \in \mathcal{X}$. We define $\mathrm{div}\phi(x) = \mathrm{Tr}J\phi(x)$. The Hilbert–Schmidt and operator norm of a matrix are denoted as $\| \cdot \|_{\mathrm{HS}}$ and $\| \cdot \|_{\mathrm{op}}$.

### 2.1 WASSERSTEIN SPACE AND CONTINUITY EQUATION

Here we summarize some of the basics of optimal transport that underlie our analysis. We denote the set of probability measures on $\mathcal{X}$ with finite second moments as $\mathcal{P}_2(\mathcal{X})$. For any $\mu \in \mathcal{P}_2(\mathcal{X})$, we express the set of measurable functions $f : \mathcal{X} \to \mathcal{X}$ with $\int \|f\|^2 \mathrm{d}\mu < \infty$ as $L^2(\mu)$, with its norm and inner product as $\| \cdot \|_{L^2(\mu)}$ and $\langle \cdot, \cdot \rangle_{L^2(\mu)}$. Given a measurable map $T : \mathcal{X} \to \mathcal{X}$ and $\mu$, we denote the pushforward measure of $\mu$ by $T$ as $T\#\mu \in \mathcal{P}_2(\mathcal{X})$, which is characterized by $\int \phi(T(x))\mathrm{d}\mu(x) = \int \phi(y)\mathrm{d}T\#\mu(y)$ for any measurable and bounded function $\phi$. Given $\mu, \nu \in \mathcal{P}_2(\mathcal{X})$, the Wasserstein distance between $\mu$ and $\nu$ is defined as $W_2^2(\mu, \nu) = \inf_{s \in \mathcal{S}(\mu,\nu)} \int \|x - y\|^2 \mathrm{d}s(x, y)$, where $\mathcal{S}(\mu, \nu)$ is the set of couplings between $\mu$ and $\nu$. This distance defines a metric on $\mathcal{P}_2(\mathcal{X})$, making $(\mathcal{P}_2(\mathcal{X}), W_2)$ the Wasserstein space, which is complete and separable.

Now we introduce a *continuous equation*. Let $T > 0$ and consider a weakly continuous map $\mu : (0, T) \to \mathcal{P}_2(\mathcal{X})$, $t \mapsto \mu_t$. The family $(\mu_t)_{t \in (0,T)}$ satisfies a continuity equation if there exists $(v_t)_{t \in (0,T)}$ such that $v_t \in L^2(\mu_t)$ and $\frac{\partial \mu_t}{\partial t} + \mathrm{div}(\mu_t v_t) = 0$ holds in the distribution sense (see Appendix B.1 for the formal meaning of *distribution sense*). A family $(\mu_t)_{t \in (0,T)}$ that satisfies a continuity equation with integrable $\|v_t\|_{L_2(\mu_t)}$ over $(0, T)$ is referred to as *absolutely continuous*. Conversely, one can construct an absolutely continuous $(\mu_t)_{t \in (0,T)}$ by selecting $(v_t)_{t \in (0,T)}$ such that they meet the above condition.

While the Wasserstein space does not inherently possess the characteristics of a Riemannian manifold, it can be endowed with a Riemannian structure and interpretation (Otto, 2001). In this in-

terpretation, the tangent space of $\mathcal{P}_2(\mathcal{X})$ at $\mu_t$, denoted as $\mathcal{T}_{\mu_t}\mathcal{P}_2(\mathcal{X})$, forms a subset of $L^2(\mu_t)$. When considering all possible $(v_t)_{t\in(0,T)}$, we call $v_t$ that exhibits the minimal $L^2(\mu_t)$ norm as the velocity field of $(\mu_t)_{t\in(0,T)}$. This minimality condition can be characterized by the requirement that $v_t \in \mathcal{T}_{\mu_t}\mathcal{P}_2(\mathcal{X})(\subset L^2(\mu_t))$.

## 2.2 SAMPLING-BASED APPROXIMATION VIA GRADIENT FLOW OF KL DIVERGENCE

We aim to obtain samples from the density $\pi(x) \propto e^{-V(x)}$ in $\mathcal{P}_2(\mathcal{X})$ under the following assumption for the potential function $V : \mathcal{X} \to \mathbb{R}$.

**Assumption 1.** *The Hessian of $V \in C^2(\mathcal{X})$, $H_V$, satisfies $\|H_V\|_{\mathrm{op}} \leq L$.*

This task can be formulated as the optimization problem over a functional space, i.e., minimizing a functional, KL divergence of $\mu$ from $\pi$ defined on Wasserstein space, that is,

$$\min_{\mu \in \mathcal{P}_2(\mathcal{X})} \mathrm{KL}(\mu|\pi), \quad \mathrm{KL}(\mu|\pi) := \int \log \frac{\mathrm{d}\mu}{\mathrm{d}\pi}(x)\mathrm{d}\mu(x), \tag{1}$$

where $\mathrm{KL}(\cdot|\pi) : \mathcal{P}_2(\mathcal{X}) \to [0, +\infty)$, $\mu \mapsto \mathrm{KL}(\mu|\pi)$ and $\mu$ is absolutely continuous with respect to (w.r.t.) $\pi$. Thus, Radon–Nikodym [1] derivative $\mathrm{d}\mu/\mathrm{d}\pi$ is available ($\mathrm{KL}(\mu|\pi) = +\infty$ otherwise).

As a method for solving Eq. (1), a gradient-descent-like algorithm utilizing the differential structure of the Wasserstein space and continuous equations (see Section 2.1) is often employed. Let the Wasserstein gradient of $\mathrm{KL}(\mu|\pi)$ at $\mu$ be $\nabla_{W_2}\mathrm{KL}(\mu|\pi)$ (the formal definition is presented in Appendix B.1). We then consider how $\mathrm{KL}(\mu|\pi)$ evolves by the continuity equation, i.e.,

$$\frac{\mathrm{d}}{\mathrm{d}t}\mathrm{KL}(\mu_t|\pi) = \langle \nabla_{W_2}\mathrm{KL}(\mu_t|\pi), v_t \rangle_{L^2(\mu_t)}, \tag{2}$$

which shows that $\mathrm{KL}(\mu|\pi)$ is minimized by choosing $v_t$ such that $\langle \nabla_{W_2}\mathrm{KL}(\mu_t|\pi), v_t \rangle_{L^2(\mu_t)} \leq 0$ and using the continuity equation. A natural choice is to use the Wasserstein gradient itself as $v_t = -\nabla_{W_2}\mathrm{KL}(\mu_t|\pi)$, which results in $\frac{\mathrm{d}}{\mathrm{d}t}\mathrm{KL}(\mu|\pi) = -\|\nabla_{W_2}\mathrm{KL}(\mu_t|\pi)\|^2_{L^2(\mu_t)} \leq 0$. According to the fact that the Wasserstein gradient of KL divergence is obtained as $\nabla_{W_2}\mathrm{KL}(\mu|\pi) = \nabla \log \frac{\mu}{\pi} \in L^2(\mu)$ (Ambrosio et al., 2005), we have

$$\frac{\mathrm{d}}{\mathrm{d}t}\mathrm{KL}(\mu_t|\pi) = -\left\|\nabla \log \frac{\mu_t}{\pi}\right\|^2_{L^2(\mu_t)}. \tag{3}$$

Many existing studies analyzed Eq. (3) under the following assumption (Bakry et al., 2013).

**Assumption 2.** *We say that the target distribution $\pi$ satisfies the LSI, if, for any $\mu \in \mathcal{P}_2(\mathcal{X})$, there exists a positive constant $C_{\mathrm{LS}}$ such that*

$$\mathrm{KL}(\mu|\pi) \leq \frac{1}{C_{\mathrm{LS}}}\left\|\nabla \log \frac{\mu}{\pi}\right\|^2_{L^2(\mu)}. \tag{4}$$

With the above inequality and Eq. (3), we have $\mathrm{KL}(\mu_t|\pi) \leq e^{-C_{\mathrm{LS}}t}\mathrm{KL}(\mu_0|\pi)$, which implies linear convergence. However, it is difficult to deal with the continuous-time equation of Eq. (3), and thus discretization such as a forward Euler discretization (Ambrosio et al., 2005) is often used. This recursion is given by

$$\mu_{t+1} = \left(I - \gamma_t \nabla \log \frac{\mu_t}{\pi}\right)\#\mu_t, \tag{5}$$

at each iteration $t$ [2], where $\gamma_t > 0$ is a stepsize and $I$ is the identity map.

---

[1] Suppose that $\mu$ is absolutely continuous w.r.t. $\pi$, i.e., $\mu \ll \pi$. Then, there exists a function $f$ such that, for any measurable set $A$, $\mu(A) = \int_A f(x)\mathrm{d}\pi(x)$. This function $f$ is referred to as the Radon–Nikodym derivative of $\mu$ w.r.t. $\pi$, denoted by $f = \mathrm{d}\mu/\mathrm{d}\pi$ (Durrett, 2019).

[2] For the sake of readability, we adopt $t$ to express both continuous and discrete time.

## 2.3 STEIN VARIATIONAL GRADIENT DESCENT

Performing optimization based on Eq. (5) is still difficult because $\mu$ is often intractable and thus $\nabla \log \frac{\mu}{\pi}$ is hard to compute. SVGD is one of the alternative gradient flow approaches to avoid this issue by projecting $\nabla \log \frac{\mu}{\pi}$ into the reproducing kernel Hilbert space (RKHS) by a kernel function.

Here, we briefly summarize the fundamental operations on the RKHS. Let $k : \mathcal{X} \times \mathcal{X} \to \mathbb{R}$ be a symmetric and positive semi-definite kernel and $\mathcal{H}_0$ be its corresponding RKHS of real-valued functions $\mathcal{X} \to \mathbb{R}$. The inner product within $\mathcal{H}_0$ is denoted as $\langle \cdot, \cdot \rangle_{\mathcal{H}_0}$, which satisfies $f(x) = \langle f, k(\cdot, x) \rangle_{\mathcal{H}_0}$ $(\forall f \in \mathcal{H}_0)$ by the reproducing property of $\mathcal{H}_0$. We also define $\mathcal{H}$ as the Cartesian product of $\mathcal{H}_0$, whose elements are expressed as $f = (f_1, \ldots, f_d)$ where $f_i \in \mathcal{H}_0$ for $i = 1, \ldots, d$. The inner product of $f, g \in \mathcal{H}$ is given by $\langle f, g \rangle_{\mathcal{H}} = \sum_{i=1}^d \langle f_i, g_i \rangle_{\mathcal{H}_0}$. If $\mu \in \mathcal{P}_2(\mathcal{X})$ and $\int k(x, x) \mathrm{d}\mu(x) < \infty$, the integral operator associated to $k$ and $\mu$ can be defined as $S_{\mu,k} f(x) := \int k(y, x) f(y) \mathrm{d}\mu(y)$, where $S_{\mu,k} : L^2(\mu) \to \mathcal{H}$ and thus $\mathcal{H} \subset L^2(\mu)$ [3]. We further define the inclusion map as $\iota : \mathcal{H} \to L^2(\mu)$, which is the adjoint of $S_{\mu,k}$. Under the map $\iota$, for $f \in L^2(\mu)$ and $g \in \mathcal{H}$, we have $\langle f, \iota g \rangle_{L^2(\mu)} = \langle \iota^* f, g \rangle_{\mathcal{H}} = \langle S_{\mu,k} f, g \rangle_{\mathcal{H}}$, where $\iota^*$ is the adjoint of $\iota$. We finally define the mapping function $P_{\mu,k} : L^2(\mu) \to L^2(\mu)$, where $P_{\mu,k} = \iota S_{\mu,k}$.

In SVGD, instead of using the Wasserstein gradient $\nabla \log \frac{\mu}{\pi}$, we employ $-P_{\mu,k} \nabla \log \frac{\mu}{\pi}$ as $v_t$ in Eq. (2), leading to the following discretized dynamics:

$$\mu_{t+1} = \left( I - \gamma_t P_{\mu,k} \nabla \log \frac{\mu_t}{\pi} \right) \# \mu_t. \tag{6}$$

The difference from Eq. (5) is that $\nabla \log \frac{\mu}{\pi}$ is mapped by $P_{\mu,k}$. If a kernel function satisfies $\lim_{\|x\| \to \infty} k(x, \cdot) \pi(x) = 0$, by using an integration by parts (Liu, 2017), we can obtain $P_{\mu,k} \nabla \log \frac{\mu}{\pi}(x) := -\int [\nabla \log \pi(y) k(y, x) + \nabla_y k(y, x)] \mathrm{d}\mu(y)$. By focusing on the continuous dynamics of the KL divergence, we have

$$\frac{\mathrm{d}}{\mathrm{d}t} \mathrm{KL}(\mu_t | \pi) = -\left\langle \nabla \log \frac{\mu_t}{\pi}, P_{\mu_t,k} \nabla \log \frac{\mu_t}{\pi} \right\rangle_{L^2(\mu_t)} = -\left\| S_{\mu_t,k} \nabla \log \frac{\mu_t}{\pi} \right\|_{\mathcal{H}}^2 =: -I_{\mathrm{stein}}(\mu_t | \pi),$$

where $I_{\mathrm{stein}}(\mu_t | \pi)$ is called as the Stein–Fisher (SF) information (Duncan et al., 2023). It is known that the square root of the SF information corresponds to the KSD. Now it is tempting to consider whether the inequality similar to LSI in Eq. (4) holds for the SF information presented below:

$$\mathrm{KL}(\mu | \pi) \le c I_{\mathrm{stein}}(\mu | \pi), \tag{7}$$

where $c$ is some positive constant. If this inequality holds, the linear convergence of SVGD holds. Unfortunately, the conditions for the validity of this inequality are not as evident as in the case of LSI and Duncan et al. (2023) has shown that Eq. (7) may not hold in many practical models with kernel functions like the RBF kernel, where the tail of $\pi$ is exponential. Hence, showing the linear convergence of KL divergence in the geometry of $\mathcal{H}$ is not straightforward. We refer to Liu (2017) and Duncan et al. (2023) for a detailed discussion of the geometry of SVGD.

Recently, Salim et al. (2022) showed the descent lemma, $\mathrm{KL}(\mu_{t+1} | \pi) \le \mathrm{KL}(\mu_t | \pi) - c\gamma I_{\mathrm{stein}}(\mu_t | \pi)$ holds where $c$ is some positive constant that depends on the problem. Although we can obtain the convergence in KSD from this inequality, the convergence KSD not necessarily means the weak convergence as discussed in Gorham & Mackey (2017).

## 3 APPROXIMATE GRADIENT FLOW

Here, we introduce a new concept of approximation for the Wasserstein gradient, $(\epsilon, \delta)$-*approximate gradient flows* (AGF). We then analyze the convergence of the KL divergence under our concept.

### 3.1 $(\epsilon, \delta)$-APPROXIMATE GRADIENT FLOW

Let us assume that a gradient flow on the Wasserstein space exists, which is induced by some velocity $v_t = g_{\mu_t}(x) \in L^2(\mu_t)$ for $x \in \mathcal{X}$. Here, $g_{\mu_t}(x)$ represents a function of $x$ only depending on $\mu_t$.

---

[3] We introduce $S_{\mu,k}$ for vector inputs $f = (f_1, \ldots, f_d)$. When $f$ is a scalar $(d = 1)$, for simplicity, we consider $S_{\mu,k}$ to be defined as applied to a single element, i.e., $S_{\mu,k} : L_0^2(\mu) \to \mathcal{H}_0$, allowing us to abuse the notation, where $L_0^2(\mu)$ is the set of a measurable function $f_1 : \mathcal{X} \to \mathbb{R}$ with $\int f_1^2 \mathrm{d}\mu < \infty$.

In the continuous-time setting, such a gradient flow is obtained via the continuity equation given as $\frac{\partial \mu_t}{\partial t} + \text{div}(\mu_t g_{\mu_t}(x)) = 0$. Under mild growth and regularity assumptions on $g_{\mu_t}(x)$ (Ambrosio et al., 2005; Bonnet & Frankowska, 2021), the existence and uniqueness of a gradient flow by $g_{\mu_t}$ is guaranteed. When considering discrete time, we assume that the recursion $\mu_{t+1} = (I - \gamma_t g_{\mu_t}) \# \mu_t$ exists, which is similar to Eq. (6).

In the presence of these, we consider the time evolution of $\text{KL}(\mu_t|\pi)$ under the velocity $v_t = g_{\mu_t}(x)$ as in Section 2.2. In the continuous-time setting, we assume that $\frac{\mathrm{d}}{\mathrm{d}t}\text{KL}(\mu_t|\pi) = \left\langle \nabla \log \frac{\mu_t}{\pi}, g_{\mu_t} \right\rangle_{L^2(\mu_t)}$. As for the discrete-time setting, we assume the following inequality with a kind of descent property:

$$\text{KL}(\mu_{t+1}|\pi) \leq \text{KL}(\mu_t|\pi) - \eta_t \left\langle \nabla \log \frac{\mu_t}{\pi}, g_{\mu_t} \right\rangle_{L^2(\mu_t)}, \tag{8}$$

where $\eta_t$ is some positive constant. Such a descent property holds both in the Wasserstein gradient flow (Ambrosio et al., 2005) and in SVGD as shown in Section 2.3.

From the above two (in)equalities, we can anticipate that when $g_{\mu_t}(x)$ exhibits behavior close to that of $\nabla \log \frac{\mu_t}{\pi}(x)$, i.e., $\left\langle \nabla \log \frac{\mu_t}{\pi}, g_{\mu_t} \right\rangle_{L^2(\mu_t)} \geq 0$ is satisfied (recall the cosine similarity in the finite-dimensional case), the KL divergence does not increase with $t$. In SVGD, for example, we set $g_{\mu_t} = P_{\mu_t,k} \nabla \log \frac{\mu_t}{\pi}$, which satisfies $\left\langle \nabla \log \frac{\mu_t}{\pi}, g_{\mu_t} \right\rangle_{L^2(\mu_t)} = I_{\text{stein}}(\mu_t|\pi) \geq 0$.

However, the condition $\left\langle \nabla \log \frac{\mu_t}{\pi}, g_{\mu_t} \right\rangle_{L^2(\mu_t)} \geq 0$ is insufficient for explicitly analyzing the convergence rate since it doesn't convey how accurate the approximation via $g_{\mu_t}$ is. To overcome this situation, we introduce a new concept of the similarity between $\nabla \log \frac{\mu_t}{\pi}(x)$ and $g_{\mu_t}(x)$ as follows.

**Definition 1.** *Suppose that $\nabla \log \frac{\mu_t}{\pi}(x) < \infty$ (a.e.) and $\|\nabla \log \frac{\mu_t}{\pi}\|_{L^2(\mu_t)} < \infty$ for all t. Then, we say a function $g_{\mu_t}(x) \in L^2(\mu_t)$ is $(\epsilon_t, \delta_t)$-AGF if the following condition holds:*

$$-\left\langle \nabla \log \frac{\mu_t}{\pi}, g_{\mu_t} \right\rangle_{L^2(\mu_t)} \leq -\epsilon_t \left\| \nabla \log \frac{\mu_t}{\pi} \right\|^2_{L^2(\mu_t)} + \delta_t, \tag{9}$$

*where $\epsilon_t, \delta_t \geq 0$.*

Eq. (9) evaluates the approximation quality of $g_{\mu_t}(x)$ for $\nabla \log \frac{\mu_t}{\pi}$ via $\{\epsilon_t, \delta_t\}$, where $\epsilon_t$ and $\delta_t$ express the relative and absolute bias of approximating $\nabla \log \frac{\mu_t}{\pi}$ by $g_{\mu_t}(x)$, respectively. This definition is motivated by the inexact gradient descent methods in finite-dimensional parameter space such as (Jaggi, 2013; Schmidt et al., 2011) and unifies some existing approximate flow methods (see Section 3.2).

Using the $(\epsilon_t, \delta_t)$-AGF, we can analyze the convergence in KL divergence qualitatively as follows.

**Lemma 1.** *Suppose that Assumption 2 is satisfied. Then, under Eq. (8), for any $T \in \mathbb{N}$, we obtain $\text{KL}(\mu_T|\pi) \leq \prod_{t=0}^{T-1}(1 - \eta_t \epsilon_t)\text{KL}(\mu_0|\pi) + \sum_{t=0}^{T-1} \delta_t \prod_{j=t+1}^{T-1}(1 - \eta_j \epsilon_j)$.*

*Proof.* By substituting Eq. (8) into Eq. (9) and applying the LSI, we obtain $\text{KL}(\mu_{t+1}|\pi) \leq (1 - \eta_t \epsilon_t)\text{KL}(\mu_t|\pi) + \delta_t$. By induction in the above, we obtain the claim. □

This lemma shows that $\epsilon_t$ and $\delta_t$ (as well as $\eta_t$) significantly impact the convergence rate.

**Remark 1.** *When $\delta_t = 0$ and $\eta_t \epsilon_t$ is independent of t, linear convergence is achieved, indicating that $g_{\mu_t}(x)$ provides a precise approximation of $\nabla \log \frac{\mu_t}{\pi}$. When $\delta = 0$ and $\eta_t \epsilon_t = \mathcal{O}(1/t^\alpha)$ with a constant $\alpha \in (0, 1]$, it indicates sub-linear convergence, which implies that the approximation quality is not so significant but it is enough to ensure the convergence in KL divergence.*

**Remark 2.** *If $\delta_t \neq 0$, the convergence is biased in terms of KL divergence. However, by employing the technique in Lee et al. (2022), it remains feasible to mitigate the impact of bias on total variation.*

## 3.2 RELATION TO EXISTING APPROXIMATE FUNCTIONAL GRADIENT FLOWS

In this section, we provide examples of AGF from existing studies. Dong et al. (2022) proposed the preconditioned functional gradient flow, where they considered approximating $\nabla \log \frac{\mu_t}{\pi}$ by neural

networks (NNs). The authors also assumed that $g_{\mu_t}(x)$, which is the output of NNs, satisfies

$$\left\| g_{\mu_t} - \nabla \log \frac{\mu_t}{\pi} \right\|_{L^2(\mu_t)}^2 \leq \epsilon \left\| \nabla \log \frac{\mu_t}{\pi} \right\|_{L^2(\mu_t)}^2, \tag{10}$$

where $\epsilon < 1$. This corresponds to a special case of our AGF with $\epsilon_t := (1-\epsilon)/2$ and $\delta_t := 0$, as confirmed by expanding the left-hand side of Eq. (10). According to Remark 1, the above inequality implies linear convergence in KL divergence. However, their method requires re-training NNs at each iteration, which yields difficulty in ensuring $\delta = 0$ in practice. Conversely, it later becomes evident that SVGD achieves $\delta_t = 0$ by using a kernel function that meets some conditions.

Lee et al. (2022; 2023) studied the score based diffusion models assuming that $g_{\mu_t}(x) = s(x) + \log \mu_t(x)$, where the $\nabla \log \pi(x)$ in the Wasserstein gradient is approximated with some measurable function $s(x)$ that satisfies

$$\left\| (s(\cdot) + \log \mu_t) - \nabla \log \frac{\mu_t}{\pi} \right\|_{L^2(\mu_t)}^2 \leq \delta. \tag{11}$$

The equation above corresponds to our AGF with $\epsilon_t = 0$, which signifies the presence of bias in the KL divergence (see Remark 2).

From the perspective of convergence analysis, the significant difference between these studies lies in the treatment of $\{\epsilon, \delta\}$. The convergence analysis in Dong et al. (2022), Lee et al. (2022), and Lee et al. (2023) assumes that $g_{\mu_t}$ achieves sufficiently small $\epsilon$ or $\delta$ according to the criteria in Eq. (10) or (11). In our study, we take the opposite approach — identifying $\{\epsilon, \delta\}$ that SVGD achieves under the AGF, and then evaluating its convergence properties.

## 4 APPLICATION TO STEIN VARIATIONAL GRADIENT DESCENT

In this section, we present the main result, the convergence of SVGD in KL divergence, obtained by applying the concept of $(\epsilon, \delta)$-AGF, and provide an overview of the proofs. Here, $\mu_t$ represents the $t$-th output of the SVGD algorithm, where $t \in \mathbb{N}$ is the number of iterations as shown in Eq. (6).

### 4.1 SUB-LINEAR CONVERGENCE OF SVGD IN KL DIVERGENCE

Our analyses are based on the following assumptions concerning the kernel function $k$.

**Assumption 3.** *The feature map $\nabla k(\cdot, x) : \mathcal{X} \to \mathcal{H}$ is continuous. Moreover, for all $x \in \mathcal{X}$, there exists $B > 0$ such that $\|k(\cdot, x)\|_{\mathcal{H}_0} \leq B$, $\sum_{i=1}^{d} \|\partial_i k(\cdot, x)\|_{\mathcal{H}_0}^2 \leq B^2$, and $\sum_{i,j=1}^{d} \|\partial_i \partial_j k(\cdot, x)\|_{\mathcal{H}_0} \leq B^2$ hold.*

**Assumption 4.** *The kernel $k$ is integrally strictly positive definite (ISPD), which means that $\int \int k(x,y) \mathrm{d}\rho(x) \mathrm{d}\rho(y) > 0$ holds for all finite nonzero signed Borel measures $\rho$.*

**Assumption 5.** *The trace of a kernel is bounded for any $\mu \in \mathcal{P}_2(\mathcal{X})$, i.e., $\int k(x,x) \mathrm{d}\mu(x) < \infty$.*

Under Assumption 5, the Hilbert–Schmidt operator $P_{\mu,k}$ has positive eigenvalues $\{\lambda_i\}$ (see Section 4.2 and Appendix C). We thus further pose the following assumption according to this fact.

**Assumption 6.** *Eigenvalues $\{\lambda_i\}$ are constant order w.r.t. $t$ and strictly positive, i.e., there exist upper and lower bounds for $\{\lambda_i\}$ that are independent of $t$ and are greater than $0$.*

Assumptions 3-6 are satisfied in the RBF kernel commonly employed in SVGD. A detailed discussion of these assumptions can be found in Appendix A.

We now show the main contribution of this paper, which establishes the sub-linear convergence of SVGD in KL divergence.

**Theorem 1.** *Suppose that Assumptions 1-6 are satisfied. Let $\alpha > 1$ and the stepsize $\gamma_t$ satisfies $\gamma_t \leq \mathcal{O}(1/t^{2/3})$ and $\gamma_t \leq (\alpha-1)\alpha B^2(1 + \|\nabla V(0) + L\mathbb{E}_\pi \|x\| + L\sqrt{2C_{\mathrm{LS}}^{-1} \mathrm{KL}(\mu_0|\pi)})(=: C_\gamma)$ for all $t$. Then, SVGD is $(c_0, 0)$-AGF and for any $T \in \mathbb{N}$, we have*

$$\mathrm{KL}(\mu_T|\pi) \leq \prod_{t=0}^{T-1} (1 - c_0 \gamma_t) \, \mathrm{KL}(\mu_0|\pi), \tag{12}$$

*where $c_0(> 0)$ is a problem-dependent constant that is independent of $t$.*

This theorem guarantees the sub-linear convergence of SVGD in KL divergence because $\lim_{t \to \infty} \frac{\mathrm{KL}(\mu_{t+1}|\pi)}{\mathrm{KL}(\mu_t|\pi)} = \lim_{t \to \infty} 1 - c_0\gamma_t = 1$. Moreover, by setting $\gamma_t = \frac{c_1}{t}$ for some positive constant $c_1$ in the above, for example, we obtain $\mathrm{KL}(\mu_T|\pi) \leq \frac{\mathrm{KL}(\mu_0|\pi)}{T^{c_1 c_0}}$.

Before outlining the proof, we position our results in comparison to existing studies. As suggested by Korba et al. (2021) and Duncan et al. (2023), it is difficult for SVGD to achieve linear convergence in KL divergence and the difficulty also surfaces in our analysis. To show our results, the step size must be $\gamma_t \leq \mathcal{O}(1/t^{2/3})$ to control $\|\nabla \log \frac{\mu_t}{\pi}\|_{L^2(\mu_t)}$, which highlights the difficulty of achieving convergence faster than sub-linear order. While Huang et al. (2023) has shown the linear convergence in a continuous-time setting, the kernel function utilized in their study is specifically designed to guarantee linear convergence and thus it is not commonly employed in practice. On the other hand, our result is established within the discrete time setting that corresponds to the SVGD algorithm, under realistic assumptions commonly met by the RBF kernel frequently adopted in SVGD.

Expanding our sight to other deterministic sampling methods based on kernel functions, sub-linear convergence has been demonstrated in the kernel herding (e.g., Chen et al. (2010); Bach et al. (2012)) and Bayesian Quadrature context (e.g., Briol et al. (2015); Futami et al. (2019)) when employing infinite-dimensional kernel functions like the RBF kernel. Our results are consistent with these facts.

## 4.2 Spectral decomposition and $(\epsilon, \delta)$-approximation

The main objective here is to provide an overview of the proof focusing on how we detect $\epsilon_t$ and $\delta_t$ in the AGF. The complete proof is in Appendix C.

To conduct analyses based on our AGF, we need to show that $\|\nabla \log \frac{\mu_t}{\pi}\|_{L^2(\mu_t)}$ is bounded for all $t$ in SVGD, which is guaranteed by the following lemma (see Appendix C.2 for complete proof).

**Lemma 2.** *Suppose that Assumptions 1-3 and 5 are satisfied. Let $\gamma_t$ satisfies $\gamma_t \leq C_\gamma$ defined in Theorem 1. Then, there exists a positive problem-dependent constant $c$ and is independent of $t$ such that, for any $t \in (0, T]$ we have $\|\nabla \log \frac{\mu_t}{\pi}\|_{L^2(\mu_t)} \leq \|\nabla \log \frac{\mu_0}{\pi}\|_{L^2(\mu_0)} + c \sum_{t=0}^{t-1} \gamma_t$.*

Now we are ready to begin the analysis of the convergence of SVGD based on AGF. Substituting $g_{\mu_t}(x) = P_{\mu_t, k} \nabla \log \frac{\mu_t}{\pi}$ into Eq. (9) and multiplying both sides by $\eta_t(> 0)$ yields

$$-\eta_t \left\langle \nabla \log \frac{\mu_t}{\pi}, P_{\mu_t, k} \nabla \log \frac{\mu_t}{\pi} \right\rangle_{L^2(\mu_t)} = -\eta_t I_{\mathrm{stein}}(\mu_t|\pi) \leq -\epsilon_t \eta_t \left\| \nabla \log \frac{\mu_t}{\pi} \right\|_{L^2(\mu_t)}^2 + \eta_t \delta_t. \tag{13}$$

According to the fact that $\eta_t I_{\mathrm{stein}}(\mu_t|\pi) \leq \mathrm{KL}(\mu_0|\pi)$ (see Appendix C), we further obtain the following inequalities:

$$\mathrm{KL}(\mu_0|\pi) \geq \eta_t I_{\mathrm{stein}}(\mu_t|\pi) \geq \epsilon_t \eta_t \left\| \nabla \log \frac{\mu_t}{\pi} \right\|_{L^2(\mu_t)}^2 - \eta_t \delta_t. \tag{14}$$

Therefore, our goal is to guarantee the existence of the above inequality. If Eq. (14) exists, we can qualitatively analyze the convergence in KL divergence by specifying $\{\epsilon_t, \delta_t\}$ and utilizing the property of AGF shown in Lemma 1 and Remarks 1 and 2.

To focus on the discussion for detecting $\{\epsilon, \delta\}$, we first mention the necessary conditions for the existence of Eq. (14) w.r.t. $\eta_t$ under our final results. As can be seen from Theorem 1, we obtain $\epsilon_t = c_0$ and $\delta_t = 0$ through the proof that we explain later, where $c_0$ is independent of $t$. In this case, from Eq. (13), it is necessary for $\eta_t \|\nabla \log \frac{\mu_t}{\pi}\|_{L^2(\mu_t)}^2$ to be uniformly upper bounded w.r.t. $t$ to compensate for the convergence based on AGF. This condition can be satisfied by setting $\eta_t$ such that it fulfills $\gamma_t \leq \mathcal{O}(1/t^{2/3})$ from Lemma 2 (see Appendix C for this derivation).

Our strategy is to show the boundedness of the following equality expressed as

$$\epsilon_t \eta_t \left\| \nabla \log \frac{\mu_t}{\pi} \right\|_{L^2(\mu_t)}^2 - \eta_t I_{\mathrm{stein}}(\mu_t|\pi) = \eta_t \left\langle \nabla \log \frac{\mu_t}{\pi}, (\epsilon_t I - P_{\mu_t, k}) \nabla \log \frac{\mu_t}{\pi} \right\rangle_{L^2(\mu_t)}. \tag{15}$$

We adopt the spectral decomposition of the kernel operator to analyze the above. Since a Hilbert–Schmidt operator $P_{\mu, k}$ is compact and self-adjoint, we have, for all $i$, $P_{\mu, k}\phi_i = \lambda_i \phi_i$, where $\phi_i \in$

Figure 1: The convergence behavior in terms of $\mathrm{KL}(\mu_T|\pi)$ and $\frac{1}{T}\sum_{t=1}^{T} I_{\mathrm{stein}}(\mu_t|\pi)$ for all $T$ under two-dimensional Gaussian distribution experiments ($\beta = 0.67 \approx 2/3$).

$L^2(\mu)$ represents an eigenfunction that satisfies a complete orthonormal system (CONS), and $\lambda_i$ is an eigenvalue corresponding to $\phi_i$. Even if these eigenvalues are ordered as $\lambda_1 \geq \lambda_2 \geq \ldots > 0$, it does not compromise generality. Moreover, the kernel function can be decomposed into $k(x,y) = \sum_{i=1}^{\infty} \lambda_i \phi_i(x)\phi_i(y)$, where the convergence of this infinite series holds in the norm of $\|\cdot\|_{L^2(\mu)}$.

Defining $v_t := \nabla \log \frac{\mu_t}{\pi}$ for simplicity in notation, we can obtain $v_t = \sum_{i=1}^{\infty} \langle v_t, \phi_i \rangle_{L^2(\mu)} \phi_i$ and $P_{\mu,k} v_t = \sum_{i=1}^{\infty} \lambda_i \langle v_t, \phi_i \rangle_{L^2(\mu)} \phi_i$ because the kernel function is dense in $L^2(\mu)$ and thus its eigenvectors are complete. We provide the discussion for non-complete eigenvectors in Appendix A. Substituting these equalities into Eq. (15), we have

$$\epsilon_t \eta_t \left\| \nabla \log \frac{\mu_t}{\pi} \right\|_{L^2(\mu_t)}^2 - \eta_t I_{\mathrm{stein}}(\mu_t|\pi) = \eta_t \sum_{i=1}^{\infty} (\epsilon_t - \lambda_i) \langle v_t, \phi_i \rangle_{L^2(\mu_t)}^2. \tag{16}$$

In the right-hand side term of the above, there exists a index $1 < j$ such that $\lambda_j > \epsilon_t > \lambda_{j+1}$ by setting sufficiently small $\epsilon_t$. Hence, by regularizing $\{\langle v_t, \phi_i \rangle_{L^2(\mu_t)}^2\}_{i=1}^{\infty}$, we can render the left-hand side of Eq. (16) negative. For that purpose, we focus on the RKHS associated with $k$ given as $\mathcal{H} = \{f \in L^2(\mu) \mid f = \sum_{i=1}^{\infty} a_i \phi_i, \sum_{i=1}^{\infty} \lambda_i^{-1} \|a_i\|^2 < \infty, a_i \in \mathbb{R}\}$, where $\mathcal{H}$ is dense in $L^2(\mu)$. In this RKHS, there exists a function $v_t^{(l)} \in \mathcal{H}$ such that the sequence of $v_t^{(l)} \to v_t$ as $l \to \infty$ in $L^2(\mu)$ norm. Thus, by approximating the original $v_t$ with $v_t^{(l)}$ in $\mathcal{H}$, we can regularize $\{\langle v_t, \phi_i \rangle_{L^2(\mu_t)}^2\}_{i=1}^{\infty}$.

Under the regularized $\{\langle v_t, \phi_i \rangle_{L^2(\mu_t)}^2\}_{i=1}^{\infty}$ in the above and sufficiently small $\epsilon_t$, we can obtain $\epsilon_t \eta_t \left\| \nabla \log \frac{\mu_t}{\pi} \right\|_{L^2(\mu_t)}^2 - \eta_t I_{\mathrm{stein}}(\mu_t|\pi) < 0$, which implies that $\delta_t = 0$ in the AGF. From Assumption 6, we can show that $\epsilon_t$ is the constant order w.r.t. $t$ and express it as $c_0$ (see Appendix C). This concludes the proof outline.

## 5 NUMERICAL EXPERIMENTS

In this section, we aim to confirm the validity of our theoretical results. We only show the results of the two-dimensional Gaussian experiments due to the page limitation. The details of the experimental settings and additional results including the Gaussian mixture can be seen in Appendix E.

We set the target distribution as the two-dimensional Gaussian distribution. We adopted the RBF kernel $k(x,y) = \exp(\frac{1}{h}\|x-x'\|_2^2)$, which is commonly used in practice and satisfies the assumptions in Section 4. The bandwidth $h$ was selected by the median trick as in Liu & Wang (2016). To appropriately verify our theoretical analysis, we simply set the decaying step size $\gamma_t = 1/(1+t^\beta)(= \mathcal{O}(1/t^\beta))$ suggested by Theorem 1 and did not use the Adagrad-based stepsize, which is adopted in related studies such as Korba et al. (2021) and others. We evaluated the KL divergence: $\mathrm{KL}(\mu_T|\pi)$ and the cumulative mean of KSD: $\frac{1}{T}\sum_{t=1}^{T} I_{\mathrm{stein}}(\mu_t|\pi)$, which are theoretically guaranteed sub-linear convergence.

**Results:** From Figures 1 and 2, we can see that SVGD with the RBF kernel tends to achieve sub-linear convergence both in $\mathrm{KL}(\mu_T|\pi)$ and in $\frac{1}{T}\sum_{t=1}^{T} I_{\mathrm{stein}}(\mu_t|\pi)$, which supports Theorem 1. As discussed in Appendix A, the bias remains in the KL divergence as we increase $T$ since we used the finite particles and thus $\delta_t \neq 0$ in AGF. Such a bias can be reduced by increasing the number of particles increases. Conversely, employing a substantial number of particles leads to

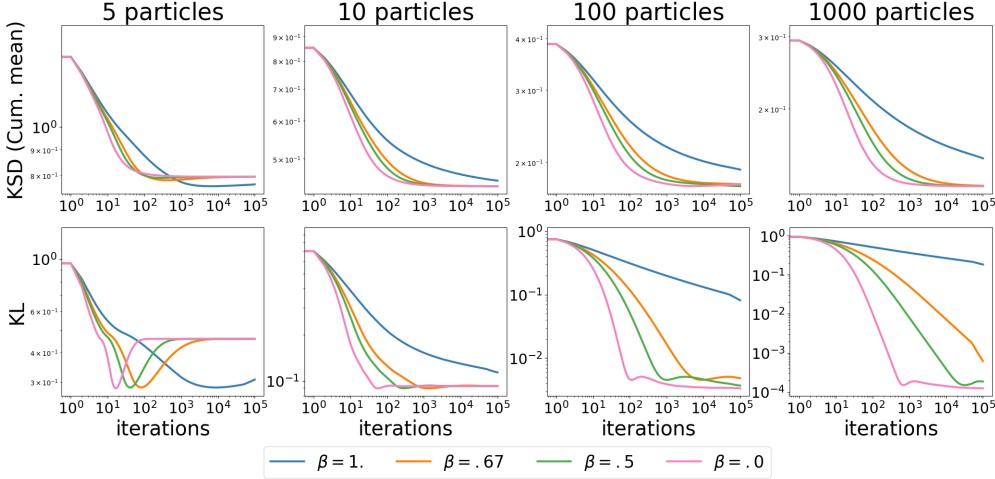

Figure 2: Convergence in $\mathrm{KL}(\mu_T|\pi)$ and $\frac{1}{T}\sum_{t=1}^{T} I_{\text{stein}}(\mu_t|\pi)$ for all $T$ under different particles and stepsize settings ($\beta = \{0., 0.5, 0.67, 1.\}$).

slower convergence for both the KSD and KL divergence. This phenomenon may be attributed to the presence of exceedingly small eigenvalues of $P_{\mu,k}$ when using a larger number of particles since the eigenvalues of the RBF kernel decay exponentially fast (Wainwright, 2019). Our numerical evaluation of eigenvalues can be seen in Appendix E. In other words, there exists a trade-off between the improvement in the approximation accuracy achieved by using a large number of particles and the convergence speed.

## 6   LIMITATION & CONCLUSION

Ensuring the convergence of SVGD in KL divergence has proven challenging in finite and infinite particle settings. Furthermore, while many studies have provided convergence guarantees for SVGD in KSD, these do not necessarily ensure its weak convergence. As a *first strategy to address this issue*, we conducted the convergence analysis of SVGD under the ideal conditions of an infinite particle setting that guarantees an accurate gradient approximation. Then, we successfully elucidated the convergence of SVGD in KL divergence in this setting. Our finding suggests weak convergence of SVGD with infinite particles, affirming its capability to approximate the expectation by the target distribution without bias, akin to MCMC.

One of limitations in our paper is the challenge in furnishing a theoretical explanation for the convergence of SVGD when employing a finite number of particles. Extending our analysis to finite particle settings using AGF is being considered as our future study. The main challenge in this extension is expected to be in determining the values of $\epsilon$ and $\delta$, primarily due to the unknown theoretical properties of gradient approximation on RKHS when dealing with correlated particles, as far as our current knowledge extends.

Another limitation is that Assumption 6 is rather strong. This assumption, introduced to ensure that $c_0$ remains of constant order w.r.t. $t$, is difficult to justify in the infinite particle setting. The pursuit of convergence guarantees grounded in milder assumptions represents a crucial avenue for future research. Furthermore, we aspire for this study to catalyze further research endeavors that aim to furnish better convergence guarantee for SVGD in KL divergence.

**Ethics Statement:** Since our paper is fundamental research based on theory, we believe that it does not cause any potential harm, societal impact, or potentially harmful consequences.

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
