# OpenReview forum: "Convergence of SVGD in KL divergence via approximate gradient flow"
_ICLR.cc/2024/Conference — Submitted to ICLR 2024_

### Official Review · Reviewer_t4TL · 2023-10-25

**Soundness:** 2 fair
**Presentation:** 2 fair
**Contribution:** 2 fair
**Rating:** 5
**Confidence:** 4

**Summary:**

This article introduces a new technique, \((\epsilon, \delta)\)-approximate gradient flow, to prove the convergence of SVGD under KL-divergence.

**Strengths:**

The convergence of SVGD under KL-divergence has always been an open question, making it a topic of great interest for theoretical researchers.

**Weaknesses:**

Assumption 6 is too strong, essentially assuming the properties of the $\nabla\log  p_t $ gradient directly. It determines the upper and lower bound for the time-dependent gradient and approximation, which almost brings the convergence.

**Questions:**

For Theorem 1, $ c_0 $ could be the most crucial constant describing the system's properties. Further explanation from the authors is hoped for.

Regarding the discrete algorithm, the $ \nabla \log p_t $ term is obviously affected by iterations, which is considered the most significant challenge. The article seems to circumvent this issue with Assumption 6, necessitating more justifications.

How to explain the non-monotonic decrease of KL in Figure 2?

---

> ### Author Response · Authors · 2023-11-18
> **Author response**
>
> ## **Answer to weakness points and questions**
>
> - **For weakness and the 2nd question:** We agree that Assumption 6 is strong, which is one of the limitations of our study.
> In the present study, this assumption was adopted because the priority is to first provide a convergence guarantee of SVGD in KL divergence. Pursuing convergence guarantees grounded in milder assumptions represents a crucial avenue for future research. We also believe that, even with such a strong assumption, our results will encourage research on guaranteed convergence under more realistic assumptions by making them available to the public.
>
> - **To the 1st question:** Regrettably, our AGF-based analysis only informs us of the existence of a time-independent constant $c_{0}$.
> A more intricate analysis is requisite to ascertain the qualitative properties of this constant. However, such an investigation exceeds the scope of this paper, as our primary focus is to ensure sub-linear convergence in KL divergence.
> Nonetheless, based on your advice, we will consider exploring the detailed nature of this constant for a comprehensive understanding of SVGD convergence in our future work.
>
> - **To the 3rd question:**  As suggested in works like [Liu et al. (2018)], there is a possibility of exhibiting oscillatory behavior around the fixed point in the setting with finite particles. We hypothesize that a similar phenomenon may have occurred in our experiment. In the SVGD context, much is still unknown about the convergence of SVGD with finite particles in the KL divergence, partly because this paper is the first proof for the infinite particle setting.
>
> ## Reference:
>
> [Liu & Wang (2018)]: Q. Liu and D. Wang. Stein Variational Gradient Descent as Moment Matching. In Advances in Neural Information Processing Systems, volume 31, pp. 8868–8877, 2018.

---

### Official Review · Reviewer_hQF6 · 2023-10-26

**Soundness:** 2 fair
**Presentation:** 3 good
**Contribution:** 2 fair
**Rating:** 3
**Confidence:** 3

**Summary:**

The authors construct a novel framework AGF to analyze convergence of SVGD in KL divergence. Based on this framework, the authors show SVGD can converge sublinearly in KL divergence.

**Strengths:**

The AGF framework can include the most existing analysis of approximate function gradient flow. It is also promising to get better results of SVGD based on this framework. Numerical experiments seem to support the theoretical results.

**Weaknesses:**

1. The key assumption is not clear and hard to verify. And I think there are some mistakes in the proof (see Question).
2. The authors do not really use AGF framework in the analysis of SVGD since $\\delta$ is always 0.
3. Basically, the method is to prove that $I\_{stein} (\\mu\_t | \\pi) \\geq c\_0 \\|\\nabla \\log\\frac{\\mu\_t}{\\pi}\\|\^2_{L^2(\\mu\_t)}$ for some positive constant $c$ and apply LSI. The contribution is not significant enough as an ICLR paper.

**Questions:**

1. In Assumption 6, $\\{\lambda_i\\}$ have positive lower bounds independent of $t$. However, Assumption 5 implies the kernel has finite trace and $\sum_i \lambda_i< \infty$. This will contradict Assumption 6. How should Assumption 6 be understood here?
2. In the proof of Theorem 1, the authors claim that "m satisfying Eq. (65) does not monotonically increase w.r.t. the iteration t" and "This allows us to identify a largest m as m′ that does not depend on t". I don't think the first claim can imply the second one since m can be arbitrarily large even if not monotonically increasing.
3. If it indeed holds that $I\_{stein} (\\mu\_t | \\pi) \\geq c\_0 \\|\\nabla \\log\\frac{\\mu\_t}{\\pi}\\|\^2_{L^2(\\mu\_t)}$, then by LSI, $I\_{stein} (\\mu\_t | \\pi) \\geq c KL(\mu_t | \pi)$ for some positive constant $c$. This seems just eq (7), which is not valid in most practical cases by Duncan et al. (2023). What is the difference between the authors'claims and Duncan et al. from this perspective?

---

> ### Author Response · Authors · 2023-11-18
> **Author response**
>
> ## **Answer to weakness points and questions**
>
> - **For weaknesses 2-3, and question 3:**
>   - The assertion that $\delta=0$ in our AGF is not readily evident from existing knowledge, such as LSI. Instead, it is a theoretical result that becomes apparent through analyzing the gradient flow of SVGD under our AGF, employing spectral decomposition and others. To commence, as [Duncan et al. (2023)] noted, conducting convergence analysis with (stein) LSI on RKHS has proven challenging. In this context, we have shown that we can evaluate convergence through "LSI-like" inequalities by demonstrating that SVGD, under the infinite particle setting, achieves $\delta=0$ in our introduced AGF. Utilizing this finding, we show the sub-linear convergence of SVGD in KL divergence. All our theoretical analysis results are non-trivial and can be attained by introducing our AGF rather than relying solely on conventional LSI.
>   - Further discussion is needed to elucidate the connection between Eq. (7) outlined in [Duncan et al. (2023)] and our findings.
> Indeed, our results are tied to Eq. (7); however, [Duncan et al. (2023)] assert that satisfying Eq. (7) for **ANY** distribution $\mu$ is challenging.
> While speculative, the distribution $\mu$ may satisfy such inequality within a context that aligns with the assumptions discussed in our paper.
> The demonstration by [Liu et al. (2023)] of linear convergence in the KL divergence for SVGD, with the constrained assumption that the target distribution is Gaussian, may imply that Eq. (7) might be valid under specific and restricted conditions.
>
> - **For question 1 (related to weakness 1):**
>   - Assumptions 5 and 6 do not contradict. Assumption 6 states that each eigenvalue $\lambda_{i}$ (for all $i$) has a positive upper and lower bound **independent of $t$**.
> Let us focus on the RBF kernel $k_{\mathrm{RBF}}$ for simplicity.
> In this case, it is known that $\lambda_{i} = \mathcal{O}(e^{-i \log i})$, thereby establishing $\lambda_{i} = \Omega(e^{-i \log i})$ (refer to Example 12.25 in [Wainwright, 2019]). This fact verifies the fulfillment of Assumption 6, as $\lambda_{i}$ is a constant order w.r.t. $t$.
> Additionally, given that the RBF kernel satisfies all conditions of Mercer's theorem (refer to Section 12.3 in [Wainwright, 2019]), we can employ the Mercer expansion for $k_{\mathrm{RBF}}$ to derive $k_{\mathrm{RBF}}(x,x) = \sum_{i=1}^{\infty} \lambda_{i}\phi_{i}^{2} < \infty$. This observation confirms the validity of Assumption 5.
>
> - **For question 2 (related to weakness 1):**  Thank you for taking the time to peruse the proofs. This concern is similar to reviewer Spa3's Q.2, so please see our response to it (please see p.20 of the modified version of this paper).
>
> ## Reference:
>
> [Duncan et al. (2023)]: A. Duncan, N. N ̈usken, and L. Szpruch. On the geometry of Stein variational gradient descent.
> Journal of Machine Learning Research, 24(56):1–39, 2023.
>
> [Liu et al. (2023)]: T. Liu, P. Ghosal, K. Balasubramanian, and N. S. Pillai. Towards understanding the dynamics of Gaussian-Stein variational gradient descent. arXiv preprint arXiv:2305.14076, 2023.
>
> [Wainwright, 2019]: M. J. Wainwright. High-Dimensional Statistics: A Non-Asymptotic Viewpoint. Cambridge Series in Statistical and Probabilistic Mathematics. Cambridge University Press, 2019.

---

> > ### Comment · Reviewer_hQF6 · 2023-11-23
> >
> > Thanks for the response and clarification of assumption 6. I would like to keep my score since AGF with $\delta_t=0$ together with LSI is stronger than equation 7, and it is not clear when this condition would be satisfied.

---

### Official Review · Reviewer_Spa3 · 2023-10-28

**Soundness:** 4 excellent
**Presentation:** 3 good
**Contribution:** 3 good
**Rating:** 6
**Confidence:** 4

**Summary:**

This paper studies the convergence of the Stein variational gradient decent(SVGD) in the population limit. Assuming the target density satisfies the log-Sobolev inequality(LSI) and other assumptions on the kernel, the authors show the sub-linear convergence of SVGD population limit in KL-divergence, which is the first KL-divergence convergence result of SVGD population limit assuming LSI. Last, numerical experiments, employing RBF kernel, are provided in Section 5 and Appendix D to verify the sublinear convergence result.

**Strengths:**

1. It is a paper with enough novelty. It combines existing results in SVGD and RKHS and prove the first convergence result of SVGD population limit in KL-divergence assuming LSI.
2. The idea and proof techniques introduced in the paper is useful for analysis on other sampling algorithms. The idea of approximate gradient flow could be used in a large class of sampling algorithms based on Wasserstein gradient flow. The proof techniques related to RKHS could be used to study other variants of SVGD as well.

**Weaknesses:**

1. The convergence result in this paper doesn't apply to the SVGD algorithm. I understand that the convergence of the finite-particle SVGD in KL-divergence is an open problem. But I am wondering that whether the idea and results in this paper can help to understand convergence of finite-particle SVGD.

**Questions:**

Questions:
1. In the first paragraph on page 1, what does the sentence ``While MCMC guarantees asymptotically obtaining unbiassed samples from the target distribution, it suffers from inefficiency due to the randomness'' mean?
2. The proof argument between $(66)$ and $(67)$ is essential, but it is a little hard to understand. I have the following questions related to the proof:

     (1) why is it enough to demonstrate that $m$ satisfying $(65)$ doesn't monotonically increase w.r.t. $t$? Even if $m$ is not monotonically increase w.r.t. $t$, $m$ could go to infinity as $t$ increases. Therefore, I think the argument should be ``it is enough to demonstrate that $m$ satisfying $(65)$ doesn't monotonically increase w.r.t. any subsequence of $t$''.

     (2) I don't understand the argument `` If $m$ were to monotonically increase, the coefficient of $(b_i^{(l)})^2$ with larger indices would also increase''. How do we prove it from $(65)$?
It would be great if a more detailed proof could be added.

Comments:

1. Although this paper focuses on convergence results for the infinite-particle SVGD, convergence results for finite-particle SVGD should be introduced in Section 1. For related results, I refer to the following two papers:

          [1] Shi, Jiaxin, and Lester Mackey. "A finite-particle convergence rate for stein variational gradient descent." arXiv preprint arXiv:2211.09721 (2022).

          [2] Liu, Tianle, et al. "Towards Understanding the Dynamics of Gaussian--Stein Variational Gradient Descent." arXiv preprint arXiv:2305.14076 (2023).

---

> ### Author Response · Authors · 2023-11-18
> **Author response**
>
> ## **For weakness**
> Your diligence in raising this crucial point is appreciated. To find our responses, kindly consult our official comments addressed to all reviewers. We have integrated a discussion of this specific matter into the newly introduced "Limitation & Conclusion" section.
>
> ## **Answers to questions**
> - **To the 1st question:** We apologize for the unclear explanation. What we would like to explain here is that MCMC tends to be computationally intensive. We modified this sentence as follows:
>   - “**While MCMC provides guarantees of producing asymptotically unbiased samples from the target density, it tends to be computationally intensive (Robert & Casella, 2004).**”
>
> - **To the 2nd question:** We agree with your opinion. We modified some sentences and explanations. It would be too complicated to describe the corrections here, so please see p.20 of the modified version of this paper.
>
> ## **Answers to comment**
> Thank you for pointing that out. We agree that [1] and [2] should have been cited. We will include the following sentences with the references of these papers in the third and fourth paragraphs of Sec. 1 for both papers:
>   - …like gradient descent. **Beyond the infinite particle setting, Shi & Mackey (2023) also showed that the SVGD with $n$ finite particles and an appropriate step size converges at the $\mathcal{O}(1/\sqrt{ \log \log n })$ order if the target distribution is sub-Gaussian with a Lipschitz score...**
>
> - …function of SVGD. **Recently, Liu et al. (2023) showed that SVGD with finite particles achieves linear convergence in KL divergence under a very limited setting where the target distribution is Gaussian. However, the analytical approach presented in previous studies makes it difficult to conduct convergence analysis based on KL divergence in a more global setting.** The reason…

---

> ### Comment · Reviewer_Spa3 · 2023-11-21
>
> Thanks for the responses. The proofs in the revised version look good to me now. I have the following a minor comment on the Assumption 6:
>
>
> 1. In [He et al., 2022]: He, Y., Balasubramanian, K., Sriperumbudur, B.K. and Lu, J., 2022. Regularized Stein variational gradient flow. arXiv preprint arXiv:2211.07861. Convergence of R-SVGD, which degenerates to SVGD when the parameter is 0, in KL-divergence was studied under assumptions of the RKHS kernel as well. Is it possible to compare their assumptions and Assumption 6 in this paper? This may provide more insights to finding the optimal kernels in SVGD.

---

> > ### Author Response · Authors · 2023-11-22
> > **Acknowlegement & our reply**
> >
> > Thank you for providing us with the important references.
> >
> > Our and [He et al., 2022]'s objectives share the same goal: providing the theoretical guarantee of the convergence in KL divergence.
> > The key to achieving this goal is deriving a lower bound for the Stein-Fisher information represented by the Fisher information such as the $(\epsilon, 0)$-AGF in Theorem 1 of our paper and $\frac{1}{2}(1-\nu)^{-1}I(\rho|\pi) \leq I_{\nu, Stein}(\rho|\pi)$ in Proposition 3 of [He et al., 2022].
> > The difference between us and [He et al., 2022] is that we achieve this goal under different assumptions for an approximate gradient flow (and a kernel function).
> > [He et al., 2022] assume the existence of the pre-image $\mathcal{J}(\rho, \pi)$ and the specific condition of the regularization parameter as in Eq. (10), which are difficult to confirm.
> > On the other hand, we derived the lower bound by strongly restricting the behavior of the eigenvalues with Assumption 6.
> > Unfortunately, it is difficult to determine which of the assumptions in both approaches is stronger; therefore, it may be challenging to discuss the optimal kernel based on these comparisons.

---

### Official Review · Reviewer_ZqiW · 2023-10-30

**Soundness:** 3 good
**Presentation:** 3 good
**Contribution:** 2 fair
**Rating:** 6
**Confidence:** 3

**Summary:**

This paper focuses on the convergence of Stein Variational Gradient Descent (SVGD) for the task of density matching when learning unknown probability densities. Specifically, the paper investigates different convergence criteria other than the Kernel Stein Discrepancy (KSD), which holds under highly restrictive assumptions, thus making it not applicable in practice. The novel criterion uses the KL divergence, the objective of SVGD, as an alternative to KSD. The core idea of the paper is to think of SVGD as an approximation of the gradient flow in the space of probability distributions. This approximation is introduced in the paper as the $(\epsilon-\delta)$ approximate gradient flow. With their new theoretical framework, the authors show that SVGD exhibits sub-linear convergence in the KL divergence.

The paper is well-written and scientifically sound. It provides a good level of preliminaries to follow the story of the paper and is complemented by some numerical results. The mathematical details are well-presented and structured. The authors often introduce and explain every step, referring to the appendix when necessary. I found the paper enjoyable to read.

However, on a more critical note, I found the paper sometimes hard to parse, perhaps due to my lack of familiarity with the topic of optimal transport at such a deep level. In general, I find the derivation reasonable and mainly easy to follow. My main issue with the current submission, however, concerns the numerical experiments (not so informative) and the take-away message one has to gather from those results.

**Strengths:**

- The paper is well written. It extensively reviews previous works and discusses how the present work fits the existing literature.
- It provides some interesting theoretical insights about the convergence of SVGD in a more relaxed setting compared to the very restrictive assumptions required by the KSD convergence analysis.
- The authors demonstrate the sublinear convergence of SVGD (Theorem 1), which represents the main result of this work.

**Weaknesses:**

- **Outlined proof is hard to grasp:** I found the proof outline (section 4.2) particularly dense and hard to follow. Perhaps having a more intuitive discussion and providing a more qualitative intuition would help people grasp the basic idea.
- **Broader impact and applicability:** I think the paper misses a discussion about the practical impact of a given study. More specifically, I’d be interested to understand how the results introduced in this work could be used when dealing with sampling tasks and approximating unnormalized target densities.
- **Conclusions are missing:** The paper does not have a conclusion section where the main findings of the work are summarized. I find this to be a substantial lack of the paper as it prevents the reader from drawing the necessary conclusion from the study.
- **Lack of numerical experiments:** While I appreciate the mathematical rigor and details of the manuscript, I perceive the numerical experiments sections (Sec. 5) to be lacking. The results seem consistent with the theory, but as far as I can tell, these are not sufficient to envision any substantial benefit from the framework presented in this work in a practical setting.
- **Take-away message from numerical experiments:** From the current version, I find it hard to understand what the central message of the numerical experiments should be.

**Questions:**

- In the introduction, the second paragraph introduces SVGD as a method to compensate for the high variance and bias from MCMC and VI methods. I am naively wondering how this would compare to other debiasing schemes often used in the context of VI, such as neural importance sampling [1]. It would be helpful if the authors could elaborate a bit on this.

- The paper always deals with the RBF kernel because it satisfies all the assumptions required for the theory to hold. However, I am wondering how easy it would be to apply the results presented in this paper to other widely adopted kernels which may not straightforwardly satisfy assumptions 1-6. Could the authors comment on this?

- I understand the paper deals with the case of infinite particles for most of the discussion. This is relevant from a theoretical standpoint, although it cannot be achieved in practice. When it comes to the final discussion, where numerical experiments are shown, the number of particles is indeed considered to be finite, which prevents the bias $\delta_t$ from vanishing. Moreover, the authors correctly state that the regime of an infinite (or even high) number of particles is practically not feasible from a computational perspective. In light of this, I wonder whether the residual bias one has to expect, resulting from the finite number of particles, can be estimated. Expanding the discussion around this point in the manuscript, I think, would be helpful.

[1] [Müller, Thomas, et al. "Neural importance sampling." ACM Transactions on Graphics (ToG) 38.5 (2019): 1-19.](https://arxiv.org/abs/1808.03856)

Minor:

- On page 3: The concept of the Radon-Nikodym derivative probably deserves a few more words. As I believe this paper could be interesting for a broad range of people who are not necessarily familiar with measure theory, I think introducing the R-N derivative with a few words or some references (perhaps just a footnote) would be helpful.

- First paragraph of page 7: “has shown the linear convergence in a continuous-time setting, the kernel function employed in **this** study is designed”. In this sentence, "this" refers to the work of Huang et al. However, the sentence may confuse the reader as it may sound that **this study** refers to the present paper. I’d recommend the authors reword this part slightly to avoid any potential confusion.

- Last paragraph before section 4.2 “[…] sampling methods based on ker […]”. I suspect *ker* might be a typo. If not, what do the authors mean by that?

- How to obtain equation (13) from equation (9) is not trivial for me, even after looking at the appendix. Perhaps adding some intuition about it would be helpful to clarify this step, as I believe it represents one of the important results of the paper.

- Below equation (16) the authors say “[…] we focus on the RKHS associated with $k$ given as $\mathcal{H}=\{…}$”. In the curly brackets, there’s a sum over k, though none of the variables in the sum actually have a subscript k. Is this meant to be a sum over $i$ instead?

- Section 4.3: kernelsatisfies -> kernel satisfies

- I found the labels in Figure 1 to be too small.

- It would be helpful to have the y-axis sharing the same range for the 2 left-most and 2 right-most plots of Figure 1. The same applies for the first and second rows of Figure 2. This would make it easier to compare visually.

---

> ### Author Response · Authors · 2023-11-18
> **Author response**
>
> ## **For weakness**
>
> - **About outlined proof:** We believe that proof outlines, as in Section 4.2, constitute a vital aspect of this paper and should be articulated rigorously, even if their presentation is somewhat dense. What kind of intuitive or qualitative explanation would facilitate a more precise understanding? We would greatly appreciate it if you could provide an example of the type of explanation you would find helpful, even if it is based on personal opinion.
> - **Broader impact and applicability:** Our study revealed that the sample obtained by SVGD converges to the target density in KL divergence, albeit exclusively in the infinite particle setting. This suggests weak convergence of SVGD with infinite particles, affirming its capability to approximate the expectation by the target distribution without bias, akin to MCMC.
> This finding could enhance users' confidence in the effectiveness and reliability of SVGD.
> In our future work, we aim to offer theoretical guarantees for the convergence of SVGD with finite particles in KL divergence, as outlined in the official comments. These discussions and official comments are summarized in a new section titled "Limitations and Conclusions."
> - **About the conclusion section:** We agree with your opinion. We have introduced a new section titled "Limitations and Conclusions," specifically addressing the content outlined in the above official comment. Please see the revised version of our paper.
> - **About lack of numerical experiments and take-away message from numerical experiments:** Our study analyzes SVGD in the infinite particle setting, acknowledging its unimplementability as an initial step towards furnishing a realistic convergence guarantee for SVGD. To corroborate the validity of the analytical results derived from the infinite particle setup through numerical experiments with finite particles, it is essential to formulate an experimental design capable of completion within a realistic computation time, even when employing a huge number of particles to simulate a near-infinite particle setup. To attain this goal, employing simple examples like the log-concave distribution experiment proves useful. We believe that the discussion of numerical experiments on convergence in more practical experimental settings will constitute a vital element in our forthcoming study on ensuring the convergence of SVGD in the KL divergence within the finite particle setting detailed in our official comment.
>
> ## **Answers to questions**
> - **To the 1st question:** MCMC faces scalability issues in handling big data, while VI efficiently approximates posterior distributions, transforming Bayesian inference into a deterministic or stochastic optimization problem. However, the accuracy of VI is influenced by the choice of the set of variational distributions, introducing bias when approximating complex target distributions with simpler ones. To reduce this bias, methods based on importance sampling have been studied; however, they pose the additional challenge that their effectiveness depends on the accuracy of the composition of the proposal distribution (e.g., $q(x; \theta)$ constructed by the neural network in [1]). In this context, SVGD was developed as a more efficient and less biased algorithm that eliminates the need for the bias-reduction methods described above by allowing the removal of constraints imposed by variational distribution sets (refer to Sec. 1 in [Liu et al., 2016] for details). To clarify this point, we modified the 1st and 2nd paragraphs of Sec. 1 as follows:
>   - In the 1st paragraph: “... **While MCMC provides guarantees of producing asymptotically unbiased samples from the target density, it tends to be computationally intensive (Robert & Casella, 2004). On the other hand, VI achieves a computationally efficient approximation of the target distribution through stochastic optimization under a simpler alternative distribution; however, it does not come with a guarantee of obtaining unbiased samples (Blei et al., 2017).**”
>   - In the top of the 2nd paragraph: “**To alleviate such sample bias while maintaining computational efficiency of VI as much as possible, Liu & Wang (2016) introduced Stein variational gradient descent (SVGD), which allows the direct approximation of the target distribution without the need for alternative distributions.**”
>
> - **To the 2nd question:** Since the RBF kernel is the most standard one employed in SVGD, many studies have adopted this kernel in the theoretical or empirical analysis (e.g., [Korba et al., 2020], [Salim et al., 2022], and [Shi et al., 2023]). We followed this context as well. In other kernels employed in SVGD, the IMQ kernel ([Shi & Mackey, 2023] focused on this kernel, for example) may be able to satisfy these assumptions.
>
> (Our responses follow in the next thread.)

---

> ### Author Response · Authors · 2023-11-18
> **Author response (2)**
>
> ## **Answers to questions (2)**
> - **To the 3rd question:** Thank you for bringing up this important point.
> For our responses, please refer to our official comments to all reviewers.
> We have incorporated a discussion of this particular point into the newly added "Limitation & Conclusion" section.
>
> ## **For MISC**
> Thank you for your careful reading. We have modified almost all the matters you pointed out. Please see the revised version of our paper.
> By labels in Fig. 1, do you mean the labels for the y-axis values in the third figure from the left? We don't consider the values between $10^{0}$ and $10^{-1}$ essential since it is sufficient to understand that the KSD value converges to smaller values as they approach the infinite particle setting.
> Concerning the y-axis range, it's important to note that KSD and KL values are incomparable. These plots aim to assess whether KSD and KL achieve sub-linear convergence in a finite particle setting. Adjusting both scales to match could result in a figure that obscures this behavior. We believe the current state provides the most intuitive means to examine sub-linear convergence.
>
> ## Reference
>
> [Liu et al., 2016]: Q. Liu and D. Wang. Stein variational gradient descent: A general purpose Bayesian inference
> algorithm. In Advances in Neural Information Processing Systems, volume 29, pp. 2378–2386, 2016.
>
> [Korba et al., 2020]: A. Korba, A. Salim, M. Arbel, G. Luise, and A. Gretton. A non-asymptotic analysis for Stein
> variational gradient descent. Advances in Neural Information Processing Systems, 33:4672–4682, 2020.
>
> [Salim et al., 2022]: A. Salim, L. Sun, and P. Richtarik. A convergence theory for SVGD in the population limit under
> Talagrand’s inequality T1. In Proceedings of the 39th International Conference on Machine Learning, volume 162, pp. 19139–19152, 2022.
>
> [Shi et al.,, 2023]: T. Liu, P. Ghosal, K. Balasubramanian, and N. S. Pillai. Towards understanding the dynamics of Gaussian-Stein variational gradient descent. arXiv preprint arXiv:2305.14076, 2023.
>
> [Shi & Mackey, 2023]: J. Shi and L. Mackey. A finite-particle convergence rate for Stein variational gradient descent. arXiv
> preprint arXiv:2211.09721, 2023.

---

> ### Comment · Reviewer_ZqiW · 2023-11-22
> **Reply to Authors**
>
> I've read the updated version of the manuscript. I acknowledge the effort of the authors in addressing all my concerns.
> For this reason, I decided to increase my score.
>
> I think the paper is much clearer now. Unfortunately, I got very sick over the past five days and did not come up with a satisfying idea for an outlined proof. Nevertheless, I believe the paper already benefitted from the improvements suggested by the referees.
>
> As far as the labels in Fig. 1 are concerned, I meant *all* the labels were a bit too small, in my opinion. I think they may be hard to read for the general public without zooming in. I understand that the values between $10^0$ and $10^{-1}$ are not relevant for the analysis, though this was not my main concern.

---

> > ### Author Response · Authors · 2023-11-22
> > **Acknowlegement**
> >
> > Thank you for your thoughtful review and the improvement in the score. We also appreciate your feedback on the size of the labels in Figure 1. In the next revised manuscript, we will make the necessary adjustments to the label size.
> >
> > P.S.: We are grateful you took the time to review our response despite your poor health. Please take care of yourself.

---

### Author Response · Authors · 2023-11-18
**Acknowledgment and comments on finite particle setting**

We sincerely appreciate your commitment of time amidst your demanding schedule to review our paper and offer invaluable insights.
We have carefully read all the comments and incorporated any necessary additions or corrections. We would be happy if you read the revised manuscript and our responses.
(The modified texts are highlighted in blue.)

## **The relationship between our study in the infinite particle setting and the finite particle setting**
We have verified that some reviewers have raised concerns about the applicability of our theoretical findings on infinite particle settings to those of finite particle settings.

The primary contribution of this paper lies in its successful elucidation of the convergence of SVGD in KL divergence, particularly in the context of the infinite particle setting.
Ensuring such convergence has proven challenging in finite and infinite particle settings (please see [Duncan et al., 2023]).
Furthermore, while many studies have provided convergence guarantees for SVGD in kernel Stein discrepancy (KSD), these do not necessarily ensure its weak convergence.
As a **first strategy to address this issue**, we conducted the convergence analysis of SVGD under the ideal conditions of an infinite particle setting that guarantees an accurate gradient approximation.
We believe extending our analysis to the finite particle setting is possible using AGF. While this extension has not been achieved yet due to the challenges mentioned below, we discuss the potential application of AGF in the finite particle setting, focusing on its influence on $\delta$ and $\epsilon$.

### **Influence on $\delta$ under the finite particles setting**

Our analysis based on $(\epsilon, \delta)$-AGF indicates that in the infinite particle setting, where the Reproducing Kernel Hilbert Space (RKHS) associated with the kernel function is dense in the $L^2(\mu)$ space, SVGD can precisely approximate the gradient $\nabla \log \frac{\mu_t}{\pi}$ without bias ($\delta=0$) (refer to Section 4.2).

On the other hand, we would expect $\delta \neq 0$ to result, which implies that SVGD, in this case, is a *“biased”* AGF since the RKHS is not dense in the $L^2(\mu)$ space.
To illustrate this intuitively, let us first examine the $m$ independent finite particles scenario.
In this scenario, we could evaluate $\delta$ because each of the $m$ eigenvectors, ordered by increasing eigenvalue, can be independently approximated.
Therefore, we could apply our proof approach in the infinite particle setting by utilizing the bias mitigating technique adopted by [Lee et al., 2022] even in such a biased AGF, as elaborated in Remark 2.

Due to particle correlations in SVGD in practice, the approximation error $\delta$ differs from when dealing with independent $m$ particles.
Yet, regarding our current understanding, there hasn't been any existing work that investigates RKHS approximations considering such correlations among samples. Therefore, a more comprehensive theoretical analysis of this point for RKHS is necessary.

### **Influence on $\epsilon$ under the finite particles setting**
As with $\delta$, we first consider $m$ independent particles for simplicity, and reordering the eigenvalues in ascending order doesn't affect generality.
In this case, the convergence is dominated by the smallest eigenvalue (i.e.,  $\lambda_{m}=\epsilon$), and it decreases as we increase the number of particles (this is intuitively represented in Figure 7-8).
Consequently, the convergence rate would deteriorate as particles increase, which we confirmed in our numerical experiments (please see Figures 1-4).
Nevertheless, in the context of practical SVGD, directly developing this argument proves challenging due to correlated particles.
To prove this argument, we need to explore further theoretical exploration to address the possibly unresolved question of constructing the eigenspace of the RKHS under correlated particles.

### Reference:

[Duncan et al., 2023]: A. Duncan, N. Nüsken, and L. Szpruch. On the geometry of Stein variational gradient descent.
Journal of Machine Learning Research, 24(56):1–39, 2023.

[Lee et al., 2022]: H. Lee, J. Lu, and Y. Tan. Convergence for score-based generative modeling with polynomial
complexity. Advances in Neural Information Processing Systems, 35:22870–22882, 2022.

---

### Meta-Review · Area_Chair_gM1L · 2023-12-12

**Metareview:**

The manuscript studies the convergence of Stein variational gradient flow using a framework of approximate Wasserstein gradient flow. Convergence of the KL divergence in discrete time is obtained for the mean field regime. The analysis however does not provide much insights on the practical setting of finite particles.

**Justification For Why Not Higher Score:**

While the infinite particle analysis is interesting, it is not necessarily connected to the practical application.

**Justification For Why Not Lower Score:**

N/A

---

### Decision · Program_Chairs · 2024-01-16

Reject